# Anti-Neurofascin Antibodies Associated with White Matter Diseases of the Central Nervous System: A Red Flag or a Red Herring?

**DOI:** 10.3390/brainsci12091124

**Published:** 2022-08-24

**Authors:** Navnika Gupta, Afsaneh Shirani, Lakshman Arcot Jayagopal, Ezequiel Piccione, Elizabeth Hartman, Rana Khalil Zabad

**Affiliations:** Department of Neurological Sciences, University of Nebraska Medical Center, 988440 Nebraska Medical Center, Omaha, NE 68198, USA

**Keywords:** anti-neurofascin antibodies, nodes of Ranvier, paranodes, demyelinating diseases, combined central and peripheral demyelination, multiple sclerosis, cerebrospinal fluid

## Abstract

Autoantibodies against nodal and paranodal proteins, specifically anti-neurofascin antibodies (ANFAs), have been recently described in central and peripheral nervous system demyelinating disorders. We retrospectively reviewed the charts of six individuals evaluated at our Multiple Sclerosis Program who tested positive for serum ANFAs on Western blot. We describe these patients’ clinical and diagnostic findings and attempt to identify features that might guide clinicians in checking for ANFAs. In our series, the women-to-men ratio was 2:1. At presentation, the median age was 60 years (range 30–70). The clinical presentation was pleiotropic and included incomplete transverse myelitis *(n* = 3), progressive myelopathy (*n* = 1), recurrent symmetric polyneuropathy (*n* = 1), and nonspecific neurological symptoms (*n* = 1). Atypical features prompting further workup included coexisting upper and lower motor neuron features, older age at presentation with active disease, atypical spinal cord MRI features, and unusual cerebrospinal fluid findings. The serum ANFAs panel was positive for the NF-155 isoform in five patients (IgM *n* = 2; IgG *n* = 2; both *n* = 1) and the NF-140 isoform in two (IgG *n* = 2). Larger studies are needed to assess the relevance of ANFAs in demyelinating nervous system diseases, their impact on long-term clinical outcomes, and associated therapeutic implications.

## 1. Introduction

Recently characterized antibodies are helping to better define immune-mediated neurological diseases of the central and peripheral nervous systems that were previously grouped into a single syndrome. Traditionally, there has been a separation between central and peripheral nervous system inflammatory demyelinating diseases. However, peripheral nervous system (PNS) involvement is increasingly recognized in multiple sclerosis (MS), the prototypic organ-specific autoimmune disease of the central nervous system (CNS) [1,2]. Furthermore, a growing body of literature describes combined central and peripheral demyelination (CCDP). Anti-neurofascin antibodies (ANFAs) have been associated with CCDP but have also been described primarily in peripheral and occasionally central demyelinating diseases. In peripheral demyelination, antibodies against neurofascin have been identified in the nodal and the paranodal regions [3], and these diseases are classified as nodopathy/paranodopathy.

Neurofascin (NF) is a cell adhesion molecule with different isoforms [3]. Neurofascin140, NF186, and NF155 isoforms are essential for assembling and stabilizing the nodal and paranodal domains of myelinated axons [3]. Neurofascin155 and NF186 are expressed in the mature nervous system at the paranodal and nodal regions, respectively, and play an essential role in saltatory conduction [4]. Neurofascin140 and NF186 are the neuronal isoforms associated with high-density voltage-gated sodium channel clusters at the node of Ranvier. However, NF140’s role is prominent during early embryogenesis. Its expression persists but declines as the nodal complex matures [5].

NF155 is the glial isoform expressed in paranodes in oligodendrocytes and Schwann cells [6,7]. Anti-NF antibodies (ANFAs) have been reported in multiple sclerosis (MS), chronic inflammatory demyelinating neuropathy (CIDP), Guillain–Barré syndrome (GBS), and CCDP [7,8,9,10,11,12]. Pathology studies of the autopsied brains of patients with MS demonstrated disruption of NF155 in the paranodal regions in demyelinating plaques [6]. Disrupted NF155 has been associated with changes in the locations of juxtaparanodal and nodal ion channels, which has been hypothesized to lead to defective axonal conduction [6]. Studies of experimental autoimmune encephalomyelitis in an animal model of MS identified a role for ANFAs in disease axonal pathology [7]. In vitro, ANFAs inhibit axonal conduction in a complement-dependent manner [7]. In vivo, antibodies targeting NF186 at the nodes of Ranvier result in complement deposition, axonal injury, and conduction block [7]. ANFAs are detected in the sera of 0.6–10% of patients with MS, particularly the primary progressive form (4.8%), suggesting a potential pathogenic role in the disease [7,10,13]. In peripheral demyelinating disorders, ANFAs lead to paranodal dissection, non-macrophage-mediated inflammation, impaired myelination, and axoglial detachment, resulting in impaired saltatory conduction and axonal disruption [8,10,14,15]. While their role in PNS demyelination is better delineated, their function in central demyelination is less well-defined and anticipated to be a work in progress. These antibodies have been primarily surveyed in the serum and, at times, cerebrospinal fluid (CSF) of patients with CIDP [8,9]. Here, we describe six patients with common neurological syndromes and some atypical features who tested positive for ANFAs in the serum in an attempt to explain the unusual findings. All but one were referred to our Multiple Sclerosis Center with clinical concern for MS. We highlight the clinical, laboratory, and imaging features that raised our suspicion for an alternative diagnosis. This paper introduces the critical question of the clinical relevance and impact of ANFAs in some commonly seen inflammatory demyelinating diseases of the CNS.

## 2. Materials and Methods

We retrospectively reviewed the charts of six individuals referred to the Multiple Sclerosis Program of an academic tertiary care center for evaluation who tested positive for ANFAs. We obtained demographics, clinical features, disease course and duration, and diagnostic workup data, including serum antibodies, CSF analysis, brain and spine magnetic resonance imaging (MRI), and nerve conduction studies/electromyography (NCS/EMG). We evaluated these patients for connective tissue disorders; nervous system paraneoplastic/autoimmune etiologies; nutritional deficiencies; and metabolic, infectious, and inflammatory etiologies. The patient’s physician obtained the patient’s consent and documented that in the patient’s medical record. For the deceased patient, the provider obtained spousal consent. The antibodies were assayed with Western blot at Washington University School of Medicine in Saint Louis, Missouri. This assay was the only commercially available test. Table 1 summarizes the serum workup results for these patients and highlights the abnormal findings. Cerebrospinal fluid analysis, when available, included cell count with differential, protein and glucose levels, protein immunoelectrophoresis, cytology, flow cytometry, paraneoplastic antibodies, and infections. We made a provisional diagnosis for each patient as it was unclear whether the positive ANFAs were pathogenic or innocent bystanders. Longitudinal follow-up of the patients and future studies might help answer this question. In a subset of patients, more than a single pathologic process might potentially contribute to the disease pathophysiology [16].

## 3. Case I

A 60-year-old right-handed Caucasian woman presented to an outside hospital for a one-day history of numbness and paresthesia in both palms and feet with rapid progression to severe bulbar and quadriplegic weakness, with onset three days after an upper respiratory infection (URI). On examination, temperature sensation was decreased in a stocking and glove distribution; muscle strength was equally reduced proximally and distally in all limbs, and deep tendon reflexes were absent. Extensive workup in the serum and CSF was unremarkable (Table 1 and Table 2). Brain and cervical spine MRI and head magnetic resonance angiography (MRA) were unremarkable. There were prolonged distal latencies and conduction velocity slowing in selective segments, suggestive of early demyelination on NCS. She was treated with intravenous immunoglobulins (IVIG) 2 g per kilogram divided over five days for acute inflammatory demyelinating polyneuropathy (AIDP). She rapidly deteriorated, necessitating intubation and ventilation. Repeat CSF analysis six days after symptom onset was unremarkable except for a protein of 50 mg/dL (reference 15–45 mg/dL). Repeat NCS on day seven after presentation showed absent motor and sensory conduction velocities. A second course of IVIG was administered with minimal improvement. She was discharged to rehabilitation with gastrostomy and tracheostomy tubes that were eventually reversed. She was released to home nearly a year later, wheelchair-bound and needing assistance with most activities of daily living on monthly IVIG treatment. As her muscle strength and gait gradually improved, IVIG was discontinued two years after symptom onset without worsening symptoms. Three years following the beginning of her first event, she experienced an acute decline with an inability to ambulate or feed herself for two weeks following a URI. Coronavirus disease 2019 (COVID-19) and other infections were ruled out. She was transferred to our institution for further evaluation. She had severe sensory loss, flaccid quadriplegia, and areflexia on examination.

Repeat brain and cervical spine MRI showed a subtle dorsal column T2 hyperintensity from C2 to C5 levels (Figure 1). There was no clinical history to suggest nitric oxide exposure. Findings consistent with diffuse and severe acute motor and sensory axonal neuropathy (AMSAN) were noted on NCS. The patient declined repeat CSF analysis. Due to both central (spinal cord lesion) and peripheral (AMSAN) involvement, serum ANFAs were assayed. Empirical IVIG was restarted. The patient’s respiratory status deteriorated, requiring prolonged mechanical ventilation support. She declined tracheostomy and long-term respiratory support, was extubated on comfort measures and expired. Post-mortem, acute motor neuropathy panel drawn two days after initiation of IVIG and sent out to the Washington University at Saint Louis laboratory returned positive for anti-NF155 IgM, anti-GM1 IgG, and anti-sulfatide IgG. We provisionally diagnosed the patient with myeloneuropathy and positive for ANFAs.

## 4. Case II

A 55-year-old right-handed Caucasian woman presented to the clinic for the evaluation of her demyelinating disease. Two years earlier, she had noticed a purplish discoloration and dragging of the left foot. Evaluation by orthopedic and vascular surgery, cardiology, and rheumatology were unrevealing. She developed a Lhermitte phenomenon radiating to the left leg a year later. Extensive serum workup was unremarkable (Table 1). Anticardiolipin antibodies tested for the purplish discoloration of the foot were positive for the IgM isoform. She had no other stigmata of antiphospholipid antibody (APLA) syndrome, and repeat testing for APLA syndrome was negative. There was a single CSF-restricted band but no other abnormalities on CSF analysis. Repeat CSF analysis showed eosinophils and a high neutrophil (35%) count. Brain MRI demonstrated several periventricular and juxtacortical lesions. Spinal cord MRI showed a dorsal cervicomedullary lesion with focal atrophy and a T4 thoracic cord lesion on T2/STIR cuts, both contrast-enhancing (Figure 2). She was initially diagnosed with primary progressive MS (PPMS) and APLA syndrome. On neurologic examination at our center, she had sensory loss to a pinprick in the left lower extremity in a stocking distribution and a mild-to-moderate decrease in muscle strength in the left arm and leg with pathological hyperreflexia. She had bilateral coarse intentional tremor with finger-to-finger and rebound in both upper extremities. Gait was left spastic monoparetic with a present Romberg’s sign. She had further tests (Table 1) and an NCS/EMG, which were unremarkable. The presence of eosinophils and a high neutrophil count on CSF analysis raised the suspicion of an antibody-mediated disease with complement activation, due to which we checked for ANFAs that returned positive for anti-NF140 IgG in the serum. Our working diagnosis for this patient is probable PPMS associated with positivity for ANFAs. The patient initially declined treatment with a disease-modifying therapy (DMT). Due to clinical and MRI worsening three years later, ocrelizumab was initiated at the most recent clinic follow-up.

## 5. Case III

A 60-year-old right-handed Caucasian woman was referred for the evaluation of abnormal brain MRI noted during a workup for parkinsonism. Brain MRI revealed ovoid and irregularly shaped periventricular and juxtacortical white matter lesions (Figure 3). Cervical and thoracic spine MRIs were unremarkable. The patient reported a history of right foot drop in childhood, imbalance, bladder urge, stress incontinence, constipation for ten years, bipolar disorder, hypertension, obesity, obstructive sleep apnea, prediabetes, and prior smoking. Neurological examination demonstrated an irregular postural and intentional tremor in both hands, no foot drop, and moderate difficulty with tandem gait. Serum testing (Table 1) was unremarkable except for positive anti-NF155 IgG. Cerebrospinal fluid analysis was unremarkable. Thyroid ultrasound and chest and abdomen computed tomography (CT) scan showed thyroid nodules and adrenal adenomas that were subsequently evaluated by endocrinology. The thyroid nodules were assessed to be multinodular goiter, and the adrenal adenomas were considered incidental. Right common peroneal neuropathy around the fibular head was reported on outside NCS/EMG in March 2020, one month before her evaluation at our clinic. Still, repeat EMG/NCS was normal. A provisional diagnosis of radiologically isolated syndrome with possible association with ANFAs was made. The patient has remained stable on clinical follow-up for over two years.

## 6. Case IV

A 64-year-old right-handed African American woman presented with a two-year history of leg numbness, imbalance, and a possible Lhermitte phenomenon. Extensive serum workup (Table 1) was non-contributory except for a creatine phosphokinase (CPK) of 282 U/L (reference 24–173). On CSF analysis, increased IgG index (2.6; reference 0–0.7) and synthesis rate (25.2; reference −9.9 to + 3.3 mg/day) and 5 CSF-restricted oligoclonal bands (OCBs) were noted. Brain MRI showed several T2 hyperintense lesions in the subcortical and periventricular regions, left anterior temporal lobe, and medulla. Cervical and thoracic MRI showed lesions at C3, C5–6, C7–T1, and T8–9 without contrast enhancement. On examination, she had left-sided hypoesthesia, mild left hemiparesis, generalized hyporeflexia, and an antalgic gait. Repeat serum testing (Table 1) was remarkable for CPK of 306 U/L, positive P/Q-type calcium channel antibody with a low titer, and anti-NF155 IgG. Repeat brain, cervical, and thoracic MRI demonstrated a new contrast-enhancing lesion at C3. Positron emission tomography (PET) scan and NCS/EMG were unremarkable. She was provisionally diagnosed with a CNS demyelinating disease with positivity for ANFAs and started on ocrelizumab.

## 7. Case V

A 70-year-old right-handed Caucasian man was referred for possible CNS demyelinating disease. He had developed sensory symptoms followed by progressive right leg weakness 20 years before presentation requiring the use of a cane. In addition, he had developed new sensory symptoms in his left leg six months before presentation. NCS/EMG showed a moderate-to-severe active right L5-S1 radiculopathy. On examination, hypoesthesia, mild weakness in the right leg, reduced vibration in all limbs, an antalgic gait, and pathological hyperreflexia were noted. Extensive serum workup (Table 1) was unremarkable except for positive anti-NF140 IgG, anti-NF155 IgM and anti-NF155 IgG. Brain, cervical, and thoracic MRI showed cord lesions at the cervico-medullary junction and from C4–C7 and a mild cervical nerve roots enhancement suggestive of mild inflammatory neuritis (Figure 4). Repeat NCS/EMG was normal with resolution of prior lumbosacral radiculopathy changes. A provisional diagnosis of myeloneuritis and positive for ANFAs was made. The patient declined DMT.

## 8. Case VI

A 30-year-old left-handed Caucasian man was hospitalized for subacute progressive sensory symptoms. Past medical history was significant for psoriasis and Roux-en-Y gastric bypass surgery six years prior with subsequent weight loss of 150 lbs. Two months post-bariatric surgery, he noted intermittent tingling in both calves, which progressed to involve the thighs and anterior abdomen associated with falls. Bilateral hip flexor weakness was noted on examination. The whole-spine MRI was unremarkable except for an incidental persistent central canal at C7. Serum vitamins and neuropathy panel (Table 1) were unremarkable except for hypovitaminosis B1, which was replaced. Cerebrospinal fluid analysis was remarkable for an elevated IgG index (1.07; ref: 0.0–0.67) and synthesis rate (5.1; ref: <3.3 mg/day) and ten CSF-restricted OCBs. EMG/NCS showed changes consistent with a right lumbosacral polyradiculopathy. The patient was lost to follow-up until three years later when he was evaluated at an outside hospital for right-hand tingling that gradually progressed over days to weakness. His sensory symptoms extended to the left hand in a stocking and glove distribution. Serum testing was unremarkable except for low vitamin B12. Cerebrospinal fluid analysis showed elevated IgG index and synthesis rate and more than five CSF-restricted OCBs. He was diagnosed with vitamin B12 deficiency and was discharged home on B12 supplementation but developed worsening symptoms three days after discharge. He presented to our institution for further evaluation. Neurological examination was significant for hypoesthesia in the right arm and legs.

Serum testing (Table 1) was unremarkable except for hypovitaminosis D, which was replaced. Brain MRI was normal. Spine MRI showed an enhancing cord lesion at C1. NCS/EMG showed changes consistent with chronic right L4-5 radiculopathy. He was diagnosed with partial transverse myelitis and treated with high-dose corticosteroids. On repeat MRI four months later, new T2/FLAIR non-contrast-enhancing demyelinating lesions were noted on the brain and cervical spine. He was diagnosed with relapsing-remitting MS and started on glatiramer acetate. Due to the combined central and peripheral nervous system involvement, ANFAs were checked and found to be positive for anti-NF155 IgM. He had recurrent transverse myelitis associated with contrast-enhancing lesions on the brain and spinal cord MRI, resulting in a DMT switch to ocrelizumab. Our working diagnosis for this patient is active relapsing-remitting MS and polyneuropathy due to bariatric surgery in the context of positivity for ANFAs.

**Table 2 brainsci-12-01124-t002:** Demographic data, clinical features, and diagnostic evaluation results for the six patients. Abbreviations: BL—Bilateral, CNS—Central Nervous System, CSF—Cerebrospinal Fluid, EDSS—Expanded Disability Severity Score, F—Female, FLAIR—Fluid attenuated Inversion Recovery, Gd—Gadolinium, IVIG—Intravenous Immunoglobulins, L—Left, M—Male, MRI—Magnetic Resonance Imaging, ND—not done, NF—Neurofascin, PNS—Peripheral Nervous System, R—Right, WML—White Matter Lesion.

	Patient I	Patient II	Patient III	Patient IV	Patient V	Patient VI
**Age, y/Sex**	60/F	55/F	60/F	64/F	70/M	30/M
**Initial symptoms**	PNS	CNS	CNS	CNS	CNS	PNS
**Age at CNS** **onset, y**	60	53	Unclear	62	43	30
**Age at PNS** **onset, y**	57	No PNS involvement	No PNS involvement	No PNS involvement	Uncertain	27
**Temporal gap between** **CNS and PNS onsets, y**	3	Not applicable	Not applicable	Not applicable	Uncertain	3
**Clinical** **presentation**	1st event: Acute onset sensorimotor symptoms in all limbs 2nd event: Acute onset sensorimotor symptoms in all limbs	Lhermitte’s sign, L foot weakness, fall, L foot purplish discoloration	Parkinsonism	BL numbness in a stocking distribution, gait imbalance	R leg tingling and weakness, gait imbalance, L leg tingling	1st event: BL legs tingling extending to the abdomen, leg weakness, falls 2nd event: BL arms and legs paresthesia and weakness
**EDSS**	2017: 7 2020: 9	5	2	5.5	6	2
**Serum ANFA (Western Blot)**	Anti-NF155 IgM	Anti-NF140 IgG	Anti-NF155 IgG	Anti-NF155 IgG	Anti-NF155 IgM Anti-NF155 IgG Anti-NF140 IgG	Anti-NF155 IgM
**CSF cell count and differentials**	1st event: No pleocytosis 2nd event: ND	1st CSF: No pleocytosis 2nd CSF: No pleocytosis, high neutrophils & eosinophils	No pleocytosis	No pleocytosis	ND	1st event: No pleocytosis 2nd event: No pleocytosis
**CSF protein, mg/dL**	1st event: 50 2nd event: ND	1st CSF: 34 2nd CSF: 30	43	32	ND	1st event: 26 2nd event: 26
**CSF IgG index and synthesis**	ND	Normal	Normal	Elevated	ND	1st and 2nd event: Increased
**CSF-restricted OCB**	ND	1	0	5	ND	1st and 2nd event: 10 & 5
**Brain MRI**	1st event: Nonspecific multifocal T2/ FLAIR WML 2nd event: Nonspecific multifocal T2/ FLAIR WML	Periventricular and juxtacortical lesion, dorsal medullary lesion extending to cervicomedullary junction	Multifocal T2/ FLAIR WML	1st MRI: Subcortical, periventricular, anterior temporal lobe, medullary lesions 2nd MRI: Stable	Periventricular, anterior cervico- medullary junction lesion	Periventricular and R frontal deep white matter lesions
**Spinal cord MRI**	1st event: Cervical spine degenerative joint Disease 2nd event: C2-5 dorsal column T2 hyperintensity	C2-3 and T4 T2/ FLAIR lesions	ND	1st MRI: C3, C5-6, C7-T1, and T8-9 lesions 2nd MRI: New C3 Gd-enhancing lesion	C2, C4-C7 lesion and cervical nerve root enhancement in the central cervical canal	1st MRI: Age-related L4-5 changes 2nd MRI: C1 contrast-enhancing lesion 3rd MRI: C2 and C5 lesions
**MRI Dissemination in Space by 2017 McDonald criteria**	Unfulfilled	Fulfilled	Unfulfilled	Fulfilled	Fulfilled	Fulfilled
**Treatment**	IVIG	Ocrelizumab	Not started	Ocrelizumab	Not started	Ocrelizumab

## 9. Discussion

We report six cases of CNS demyelinating diseases associated with ANFAs with or without PNS involvement. All except one patient were in their sixth decade or older. Previous studies looking at patients with neurological disorders in the presence of ANFAs reported a wide age range of 10–80 years [9,10,17]. Four out of six patients in our case series were women. We suspect that gender predominance may follow the phenotype of the disease. For instance, those patients with CIDP might be predominantly men, and those with MS-like disease might be primarily women. Age of onset may also play a role in gender dominance. Based on the literature review, it is likely that the incidence is similar in men and women [10,18,19].

Clinically, patients may develop isolated CNS or PNS disease or a combination of both. When both systems are affected, the involvement may occur simultaneously or sequentially (3 or more months apart) [10,11]. In our series, isolated CNS involvement with normal NCS/EMG was observed in cases 2, 3, and 4. The diagnosis in these patients might be PPMS, early presentation of CCDP, or a different CNS-restricted disease. In a cohort of patients (*n* = 83) with PPMS, 4.8% had ANFAs and were noted to have worse disability [12]. However, there was no information on PNS involvement or OCB status in this ANFA-seropositive group. Some of the patients reported by Stitch et al. and whom we report meet the 2017 McDonald criteria for dissemination in space on MRI. The unusual or atypical findings in our series prompted further workup in search of an alternative explanation.

Several studies have investigated the presence of serum ANFAs including one analyzing their presence in both the serum and CSF [7,8,9,10,11,12,17,18,20]. It is worth noting that these studies included patients with peripheral neuropathies without known CNS involvement. The serum ANFAs detected in these series were anti-NF140 IgG, anti-NF155 IgM, and IgG, anti-NF186 IgM, and IgG. The detected subclasses of IgG were IgG1, IgG2, IgG3, and IgG4 [8,9,17,18]. Stich et al. assayed the simultaneous presence of serum and CSF ANFAs in five patients with MS, and none had ANFAs in the CSF [12]. Studies have also examined CSF inflammatory markers in patients with positive ANFAs. Kawamura et al. (*n* = 7) reported that all the cases had elevated CSF protein, though only one had a CSF-restricted band [10]. In our series, patients were positive for serum anti-NF155 IgM and IgG, and anti-NF140 IgG. Two out of five patients had CSF-restricted OCBs. Understandably, some of these CSF-OCBs might be transient, as reported with other autoimmune inflammatory diseases of the CNS [21]. One patient with positive serum anti-NF140 IgG (patient 2) had eosinophils and a high neutrophil count on CSF analysis, raising the possibility of a complement-mediated immune attack.

Interestingly, three of our patients with both CNS and PNS involvement had anti-NF 155 IgM isoforms with or without IgG. The pathogenic significance of ANFA IgM remains elusive [20]. Interestingly, the presence of IgM oligoclonal bands in the CSF of patients with MS may be predictive of a worse outcome [22]. In the case of ANFA-positive patients, it is premature to speculate regarding the significance of IgM positivity.

Abnormal NCS/ EMG supported AMSAN (case 1) and lumbosacral polyradiculopathy (case 6). Lumbar spine MRI in case 6 was negative for neural foramina or spinal stenosis to explain an anatomic compressive etiology of lumbosacral polyradiculopathy. Thus, an atypical form of CIDP cannot be excluded [14,23]. Additionally, CIDP diagnosis by electrophysiology remains challenging, with the possibility of misdiagnosis [24]. One of our patients (case 5) had cervical peripheral nerve root involvement on spine MRI with normal NCS/EMG. The normalcy of the NCS/EMG could be secondary to chronic immune sensory polyradiculopathy (CISP) that might be difficult to diagnose [25].

Kawamura et al. described the neuroimaging features of seven patients with positive anti-NF155 IgG antibody [10]. In their study, brain MRI showed diffuse or multifocal cerebral, cerebellar, and brainstem white matter lesions, some with contrast enhancement. In addition, some patients had multifocal short segment cord lesions on spine MRI or cauda equina hypertrophy with contrast enhancement. Six patients in their study fulfilled the 2010 McDonald diagnostic criteria for relapsing–remitting MS. One patient had diffuse cerebral white matter lesions that did not fulfill these criteria. Abnormalities on NCS/EMG included prolonged distal and F-wave latencies, slow conduction velocities, conduction block, and temporal dispersion [10]. Paranodal antibodies have mostly been reported in treatment-refractory CIDP with prominent distal involvement, ataxia, and disabling tremor [26,27]. In our cases, we found acute onset demyelinating or axonal neuropathy and polyradiculopathy without demyelination. These findings expand the spectrum of peripheral nerve involvement of paranodal antibodies and could suggest that PNS manifestations are not restricted to the classic refractory distal demyelinating phenotype. In our series, patients 2–5 fulfilled the 2017 McDonald MS criteria for dissemination in space on MRI; patients 4 and 6 fulfilled the 2017 McDonald MS MRI criteria for dissemination in time. In addition, patient 5 had a longitudinally extensive lesion from C4–C7 similar to that previously reported in association with ANFAs with cervical nerve root enhancement [10].

One might argue that the positive ANFAs are of dubious significance in our series as they were all assayed with Western blot and the antibodies may reflect non-specific bindings that do not have pathological relevance. However, two earlier studies demonstrated equivalent results with enzyme-linked immunoassay (ELISA) and cell-based assay (CBA) [8,19]. In addition, another study by Kadoya et al. showed a similar frequency of ANFAs in a cohort of patients with CIDP assayed using four different techniques, including ELISA (using a recombinant human NF protein), Western blot, CBA, and immunohistochemistry [28].

We would like to highlight the features that compelled us to check for ANFAs in our patients, as these are important to clinicians. In case 1, the combined cervical dorsal column hyperintensity on MRI and severe and recurrent acute neuropathy meeting the criteria for AMSAN on NCS/EMG was key to diagnosis [29]. In particular, AMSAN is a known nodopathy/paranodopathy. The spinal dorsal column changes noted in case 1 are unlikely to reflect Wallerian degeneration secondary to AMSAN due to focal (rather than diffuse) cord involvement. In case 2, the presence of eosinophils and increased neutrophils in the CSF suggested a complement-dependent antibody-mediated disease, prompting us to look for MS mimics. Clues for further workup in case 3 included a later onset age, cardiovascular risk factors, atypical history, and mixed brain MRI findings, including non-specific subcortical and ovoid periventricular lesions associated with black holes. Finally, the late-onset presentation associated with disease activity (cases 4 and 5) and the presence of peripheral nerve root enhancement (case 5) on MRI prompted the search for ANFAs. We believe that this is a new syndrome associated with positive ANFAs. Larger and longitudinal studies are required to determine if ANFAs are transient or chronic or if there is a new subtype of PPMS associated with ANFAs.

Treatment with corticosteroids, IVIG, plasma exchange, rituximab, cyclophosphamide, and methotrexate [28] have been reported in managing isolated PNS disease or CCDP. Occasionally, interferon-beta was used with limited benefit [11]. However, there are limited data on treating patients with exclusive CNS involvement.

On a final, important note, although the literature refers to the ANFA-associated diseases as demyelinating, the immune attack targets the nodes and paranodes. Recognizing that the pathophysiology of antibody-mediated nodopathies and paranodopathies depends on the subclass of the IgG in question, these antibodies cause a conduction block as a first step. Then, demyelination ensues due to a disconnect in the axoglial interaction. These findings were demonstrated in a rat model of AIDP, where the loss of NF186 and gliomedin, another cellular adhesion molecule, initiated paranodal demyelination [30]. A subsequent paper from the same laboratory revealed a lack of demyelination in the rat model [31]. Finally, a human study by Koike et al. demonstrated the absence of inflammatory cells and onion bulb formation and mildly decreased myelin fibers in the sciatic nerve of patients affected by anti-NF 155, and anti-contactin antibodies IgG4 on light microscopy [15]. Segmental remyelination was comparable with normal controls. Electron microscopy confirmed these findings and the absence of macrophage-mediated demyelination.

Our study, nonetheless, has several limitations. First, we evaluated patients with central nervous system syndromes exclusively, as they are the referral basis for our clinic. Thus, we did not include patients with peripheral nervous system syndromes. We checked for ANFAs in these patients when we suspected an alternative diagnosis due to atypical features. Second, we tested for serum ANFAs only because a test for CSF is commercially unavailable. It is probably performed in research laboratories, but we did not investigate that further. Third, we checked for antibodies against NF-140 in patient 2 but did not check for anti-NF 186 antibodies. Allegedly, antibodies against NF-140 cross-react with anti-NF 186 antibodies making result interpretation difficult. Fourth, we checked for ANFAs by the Western blot technique only and did not perform a confirmatory testing with an alternative technique like ELISA, CBA, or teased nerve fiber preparation. We reached out to the commercial laboratory for further clarification of the assay methodology. Reportedly, ELISA was used when the test was launched, but the Western blot methodology was subsequently found to be more specific. Furthermore, this test has been validated. We acknowledge that the gold standard method is a subject of heated debate, and the answer will likely be possible with time.

## 10. Conclusions

Autoantibody-associated demyelinating or demyelinating-like diseases are a rapidly growing field of interest. New antibodies are being detected speedily due to the availability of high-quality assays. Whether ANFAs are part of a unique and emerging disease entity, disease modifiers, or inconsequential remains to be elucidated with time. If ANFAs-associated diseases represent distinct entities (similar to myelin oligodendrocyte glycoprotein associated demyelination or MOGAD), their clinical phenotypes and spectra are yet to be defined. Diagnostic criteria might be defined with a better understanding of the clinical presentations, helping clinicians make the correct diagnosis. Clinicians should maintain a high index of suspicion and consider checking for ANFAs in the presence of atypical findings such as later age of onset, atypical CSF findings, and combined CNS and PNS involvement, especially when confirmed on imaging and/or with neurophysiological studies.

## Figures and Tables

**Figure 1 brainsci-12-01124-f001:**
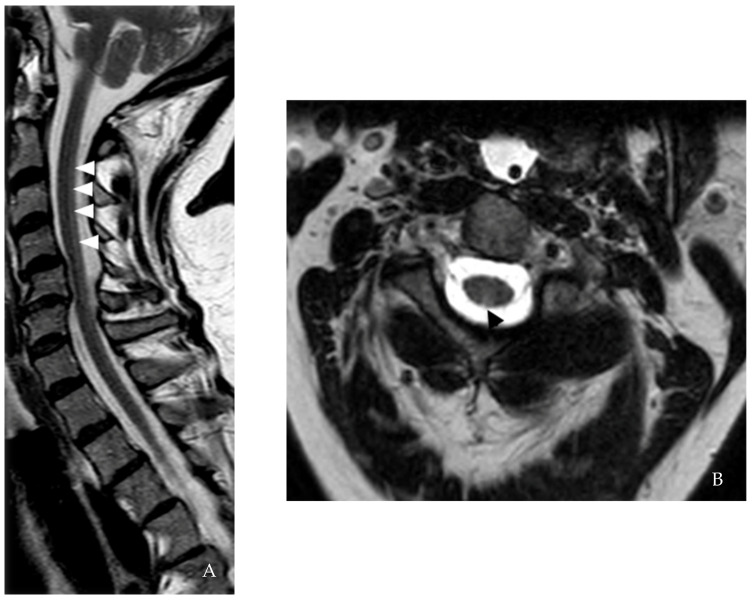
**Case I.** (**A**) Saggital STIR cervical spine MRI shows selective T2-hyperintensity of the dorsal column bilaterally from C2-5 (white arrows). (**B**) Axial T2 confirmed the presence of the lesion at the same level (black arrows).

**Figure 2 brainsci-12-01124-f002:**
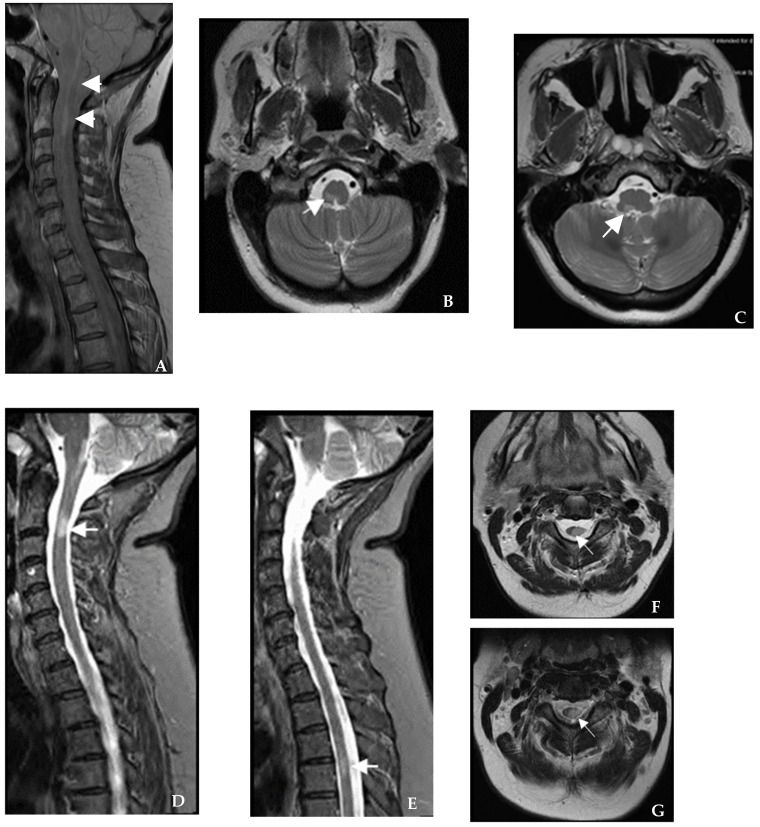
**Case II.** (**A**) Sagittal proton density brain and cervical spine MRI show an elongated lesion from the dorsal medullary area to C1–C2 (white arrows). (**B**–**C**) Axial cut at the specified level confirmed the lesions (white arrows). (**D**–**E**) Sagittal STIR cervical spine MRI shows C2–C3 and T4 cord lesions (white arrows). (**F**–**G**) Axial cut at the specified level confirmed the lesions (white arrows), and additionally showed involvement of the majority of the cord cross-section associated with left hemi-cord atrophy at C2–C4 (**F**).

**Figure 3 brainsci-12-01124-f003:**
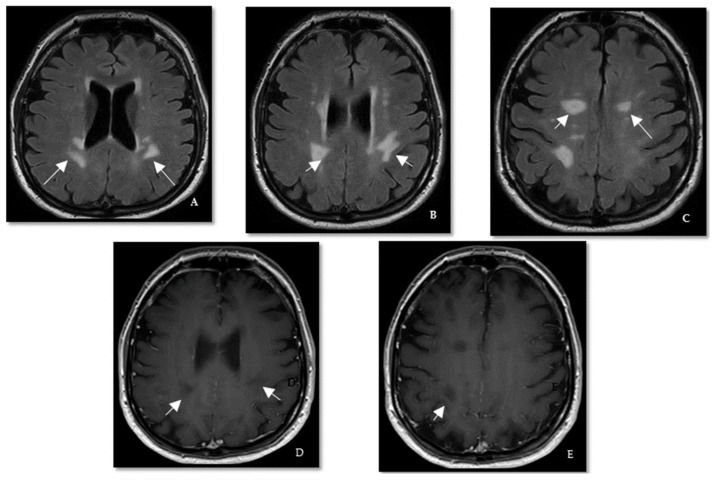
**Case III.** (**A**–**C**) Axial T2/FLAIR brain MRI shows periventricular hyperintense lesions (white arrows). (**D**–**E**) Axial T1-Gad brain MRI shows non-contrast enhancing black holes (white arrows).

**Figure 4 brainsci-12-01124-f004:**
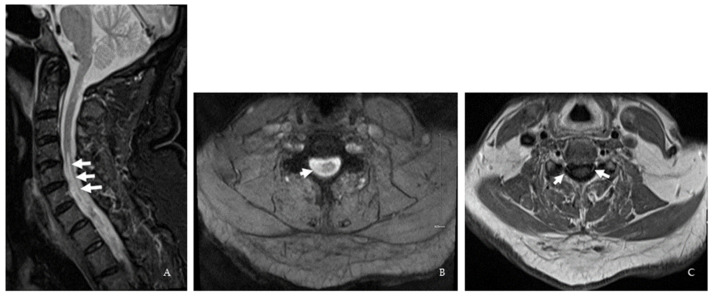
**Case V.** (**A**) Sagittal STIR cervical spine MRI shows a longitudinally extensive (5 segments) T2 hyperintensity (white arrows). (**B**) Axial cut at the specified level shows the involvement of the lateral aspect of the cervical spinal cord (white arrow). (**C**) Axial T1 post-gadolinium at the cervical level shows bilateral nerve root enhancement (white arrows).

**Table 1 brainsci-12-01124-t001:** Summary of the serum laboratory test results for the six patients. All the columns except the last one (positive findings) reflect all the tests that were performed and were unremarkable. Abbreviations: ACE-angiotensin-converting enzyme, ANA-antinuclear antibody, ANCA-antineutrophil cytoplasmic antibodies, anti-ds DNA-anti-double stranded DNA, APLA- antiphospholipid antibodies, anti-TPO-anti thyroid peroxidase, AQP-4-aquaporin-4, CK-creatine kinase, HgbA1c- hemoglobin A1c, HIV-human immunodeficiency virus, HTLV-human T-lymphotropic virus, Ig– immunoglobulins, SIIF-serum immunofixation, MMA- methylmalonic acid, *M. pneumoniae*-*Mycoplasma pneumoniae*, MOG- myelin oligodendrocyte glycoprotein, PTH-parathyroid hormone, RF-rheumatoid factor, SPEP-serum protein electrophoresis, TSH-thyroid stimulating hormone, VLCFA-very-long-chain fatty acid, VGKC-Voltage-gated potassium channel, WNV-West Nile Virus.

Test	Infectious	Autoimmune	Paraneoplastic Antibody Panel	Inflammatory	Hematology	Nutritional	Miscellaneous	Positive Findings
**Case I**	HIV, hepatitis, *M. pneunoniae* IgM, Lyme & WNV serologies	None	Negative, except for findings in the “Positive” column	ACE	SIF, SPEP	Vitamin B12, MMA, homocysteine	None	*M. pneumoniae* IgM, VGKC titer = 0.13; ref: ≤0.02 nmol/L
**Case II**	Hepatitis B, HTLV-1 & 2, Lyme, WNV & syphilis serologies	ANA, ANCA, anti-ds DNA, anti-TPO, AQP-4, RF & MOG IgG	Negative	ACE, ESR	C3-C4, SPEP, SIF	Amino acids, VLCFA, copper, ferritin, MMA, vitamins B1/B6/B12/D	HgbA1c, TSH	APLA IgM (repeat test negative)
**Case III**		ANA, ANCA, AQP-4, & MOG IgG, celiac screen	Negative	CRP	SIF, C3-C4, SPEP,	Ferritin, folic acid, vitamin B12, VLCFA	None	
**Case IV**	HIV & Lyme serologies	ANA, AQP-4, & MOG IgG, lupus anticoagulant	Negative, except for findings in the “Positive” column	CRP, ESR	None	Copper, vitamins B12/D/E, zinc	None	CPK: 306 U/L (ref: 38-234). P/Q-type calcium channel antibody
**Case V**	HTLV-1 & 2	AQP-4 & MOG IgG	Negative	None	None	Copper, folate, MMA, vitamins B12/D/E	None	None
**Case VI**	1st event: Syphilis serology 2nd event: Hepatitis, HIV & Lyme serologies	2nd event: ANCA, AQP-4 & MOG IgG	2nd event: Negative	2nd event: ACE	2nd event: Igs, SPEP,	1st event: Copper, iron panel, MMA, vitamins A/B6/ B12/D, zinc 2nd Event: Copper, homocysteine, vitamins A/B12/E	1st event: HgbA1c, PTH 2nd event: HgbA1c, TSH	1st event: Hypovitaminosis B1 2nd event: Hypovitaminosis D

## Data Availability

Not applicable.

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
