# Peer review of "Anti-Neurofascin Antibodies Associated with White Matter Diseases of the Central Nervous System: A Red Flag or a Red Herring?"

_brainsci, 2022, doi:10.3390/brainsci12091124_

Round 1
Reviewer 1 Report
The authors have reported six patients with serum anti-neurofascin antibodies. The details of cases had been well described. The discussion was comprehensive and pointed out the importance of ANFAs in some patients with atypical syndrome and combined CNS and PNS involvement. It would be better to provide some representative images of western blot bands of serum ANFAs if possible.
Author Response
Dear Reviewer,
We thank you for taking the time for reviewing our manuscript and appreciate your timely feedback. The ANFAs assay is done commercially and the Western Blot results are not available to us.
Sincerely,
Rana K. Zabad
Reviewer 2 Report
Gupta et al describe the cases of six patients with antibodies against neurofascin and CNS demyelinating disorders. Antibodies against neurofascin-155 (NF155) has been reported to be associated with combined central and peripheral demyelinating diseases (CCPD). However, very few studies have confirmed this association. Instead antibodies against NF155 or pan-neurofascin have been widely reported in chronic inflammatory demyelinating polyneuropathy (CIDP) or Guillain-Barre syndrome (GBS). Here, the authors describe the cases of six patients with CNS demyelinating disorders or CCPD associated with anti-neurofascin antibodies: IgM against NF155, IgG against NF155 or IgG against NF155 and NF140. The study is well written and the cases are well described. My main concern is that the authors assume that anti-neurofascin antibodies can be assigned into a single group: ANFAs. It is quite clear now that patients with IgG4 against NF155 or pan-neurofascin are associated with CIDP groups with distinct clinical presentations. There are also recent evidence suggesting that IgG1 against pan-neurofascin may be also associated with severe GBS forms. Concerning IgM against NF155, albeit there are some reports of anti-NF155 or NF186 IgM in GBS and CIDP patients, the clinical relevance of those is unclear.
Here, the authors mixed together the cases of patients with IgM against NF155, IgG against NF155, and patients with seemingly anti-pan-neurofascin IgG which is confusing and is complicating the interpretation and conclusion. I am also very doubtful concerning the clinical value of anti-NF155 IgM.
One of my main concern is that the authors did not test the antibody isotype/titers, and used a single unconventional method (western blot) to detect those antibodies. Since the studies is focused on the clinical description of patients with ANFAs, I think it is important that the authors validate their tests with other technique and determine the isotype and titers. I am currently not convinced that anti-NF155 IgM are responsible for the CCPD in patients I and VI.
Patient I is reactive against GM1 which is classically associated with AMSAN. Thus why suspect IgM against NF155?
Similarly, patient VI developed a polyneuropathy following gastric surgery and B1 deficiency. The association MS and polyneuropathy may be just fortunate, and not related to the presence of anti-NF55 IgM.
Concerning the antibody panel, why did the authors tested NF140 and not NF186? The authors should mention in the methods which antigens have been tested. If not done already, the authors should test for anti-NF186 IgG. This will make more sense than anti-NF140.
For instance, does Patient 2 react against NF140 only? This would be surprising knowing that the whole amino acid sequence of NF140 is present in NF155 and NF186. Thus antibodies to NF140 should recognize NF155 and NF186.
Author Response
Dear Reviewer,
We thank you for taking the time for reviewing our manuscript and appreciate your timely feedback. Please find the responses to your valuable comments below. We remain at your disposition for further questions or concerns.
Gupta et al describe the cases of six patients with antibodies against neurofascin and CNS demyelinating disorders. Antibodies against neurofascin-155 (NF155) has been reported to be associated with combined central and peripheral demyelinating diseases (CCPD). However, very few studies have confirmed this association. Instead antibodies against NF155 or pan-neurofascin have been widely reported in chronic inflammatory demyelinating polyneuropathy (CIDP) or Guillain-Barre syndrome (GBS). Here, the authors describe the cases of six patients with CNS demyelinating disorders or CCPD associated with anti-neurofascin antibodies: IgM against NF155, IgG against NF155 or IgG against NF155 and NF140. The study is well-written, and the cases are well described. My main concern is that the authors assume that anti-neurofascin antibodies can be assigned into a single group: ANFAs. It is quite clear now that patients with IgG4 against NF155 or pan-neurofascin are associated with CIDP groups with distinct clinical presentations. There are also recent evidence suggesting that IgG1 against pan-neurofascin may be also associated with severe GBS forms.
Response: We fully appreciate the author’s concern about lumping all antibodies into a single group; to this end, we have used the term ANFAs in the plural form to highlight that there are several of them. Assigning all the antibodies into a single group was made only for the ease of description and by no means is an assumption of a single pathophysiology. We remain open to any other “name” or “designation” the author might suggest.
Concerning IgM against NF155, albeit there are some reports of anti-NF155 or NF186 IgM in GBS and CIDP patients, the clinical relevance of those is unclear. Here, the authors mixed together the cases of patients with IgM against NF155, IgG against NF155, and patients with seemingly anti-pan-neurofascin IgG which is confusing and is complicating the interpretation and conclusion. I am also very doubtful concerning the clinical value of anti-NF155 IgM.
Response: Regarding the significance of Ig-M anti-Neurofascin antibodies, we agree that the clinical significance is unclear for now and the door remains open for further clarifications, if any. For now, the absence of evidence is not necessary evidence of absence.
One of my main concern is that the authors did not test the antibody isotype/titers and used a single unconventional method (western blot) to detect those antibodies. Since the studies is focused on the clinical description of patients with ANFAs, I think it is important that the authors validate their tests with other technique and determine the isotype and titers. I am currently not convinced that anti-NF155 IgM are responsible for the CCPD in patients I and VI.
Response: Regarding the Western Blot methodology, this was the only test available commercially for assaying anti-neurofascin antibodies at the time. Additionally, we have tried to address this caveat in the discussion, to the best of our abilities as stated in the discussion “One might argue that the positive ANFAs are of dubious significance in our series as they were all assayed by western blot and that the antibodies may reflect non-specific bindings which do not have pathological relevance. However, two earlier studies demonstrated equivalent results with enzyme-linked immunoassay (ELISA) and cell-based assay (CBA) 6,15. In addition, another study by Kadoya et al. showed a similar frequency of ANFAs in a cohort of patients with CIDP, assayed by four different techniques, including enzyme-linked immunosorbent assay (using a recombinant human NF protein), western blot, cell-based assay, and immunohistochemistry.”
Patient I is reactive against GM1 which is classically associated with AMSAN. Thus, why suspect IgM against NF155?
Response: We agree that anti-GM1 antibody is classically associated with AMSAN. However, since anti-GM1 antibody targets a node antigen, with the coexistence of a dorsal column T2/FLAIR lesion we thought of a nodopathy as a potential explanation of a combined CNS and PNS pathology, which prompted us to check for anti-neurofascin antibody.
Similarly, patient VI developed a polyneuropathy following gastric surgery and B1 deficiency. The association MS and polyneuropathy may be just fortunate, and not related to the presence of anti-NF55 IgM.
Response: We agree that the association between MS and polyneuropathy may be coincidental, as we pointed out in our discussion, in this case and others. However, the presence of a peripheral nervous system and central nervous system pathology in this case raised suspicion for a common pathophysiologic process affecting both the systems that prompted us to check for anti-NF155 IgM. Since, antibodies against neurofascin have been described more recently, longitudinal follow-up over time might help better characterize the underlying disease process.
Concerning the antibody panel, why did the authors test NF140 and not NF186? The authors should mention in the methods which antigens have been tested. If not done already, the authors should test for anti-NF186 IgG. This will make more sense than anti-NF140.For instance, does Patient 2 react against NF140 only? This would be surprising knowing that the whole amino acid sequence of NF140 is present in NF155 and NF186. Thus, antibodies to NF140 should recognize NF155 and NF186.
Response: We agree with the reviewer and thank them for bringing this to our attention. Not testing for NF-186 is a potential limitation of our study. We have checked with the Washington University laboratory and NF-186 antibody typically cross reacts with NF-140. This has been updated in the manuscript (page 3, line 281-285).
Reviewer 3 Report
This manuscript describes a small cohort of central and peripheral nervous system demyelinating disorders with anti-neurofascin antibodies. I find the manuscript interesting, and it is clearly written, but I have serious concerns regarding the methodology of antibody detection. Therefore, I recommend major revision.
Major points:
1. The authors based their results in western blot detection. Western blot is not a validated test for nodal/paranodal antibody screening. Furthermore, it is recommended to perform at least two techniques to confirm the positivity, because false positives have been previously described (Martín-Aguilar et al, Neurology, 2020). The recent published guidelines on diagnosis and treatment of CIDP (Van den Bergh et al. EAN/PNS Guidelines, J Peripher Nerv Syst, 2021) recommend performing a cell-based assay (CBA), ELISA or teased-nerve immunohistochemistry and always performing a confirmatory test with a second technique. In fact, the authors discussed that antibodies detected by western blot may reflect non-specific bindings without pathological relevance (discussion, line 239), but they also state that two articles demonstrated the equivalence between ELISA and CBA. These two articles demonstrate that ELISA and CBA are equal methods to detect nodo-paranodal antibodies, but not western blot. Thus, I recommend performing at least two different techniques to the western blot to confirm the positive patients in your case series.
2. In the introduction the authors state: “Anti-neurofascin-antibodies (ANFAs) might characterize an emerging group of immune-mediated neurological disorders that have not yet been fully described in terms of presentation, diagnosis, and management.” This affirmation needs clarification: in terms of peripheral nervous system, anti-neurofascin antibodies are disease-specific antibodies that have been described in a small subset of patients with CIDP sharing immunopathologic mechanisms, clinical features, and treatment response and differing from those of typical CIDP. Moreover, this has led to the appearance of the autoimmune nodopathy (AN) diagnostic category in the recent update of the European Academy of Neurology/ Peripheral Nerve Society CIDP diagnostic guidelines (Van den Bergh et al. EAN/PNS Guidelines, J Peripher Nerv Syst, 2021). On the other side, combined central and peripheral demyelination (CCPD) has been described in two cohorts: Kawamura et al. Neurology 2013 and Cortese et al., J Neurol Sci, 2016, last reference not included in the manuscript. Please, add a better explanation of neurofascin antibodies in peripheral nervous system and discuss that its value in CNS and in CCPD is not so well-known.
3. From table 1, 5 out of 6 patients have been tested for MOG/NMO antibodies. Which test have you used to test NMO/MOG antibodies? Patient 1 should be also tested for NMO/MOG, as she has a longitudinally extensive myelitis and NF155 have been reported to appear in some NMO patients (Chang et al. Clin Exp Immunol, 2021).
Minor points:
1. Final diagnosis is not clear in most of presented cases and a lot of tests and explanations for each case can be collected in a table. Consider reducing the length of the clinical cases, trying to include the results of the tests in table 2 and, in each case, try to approximate the diagnosis whenever possible.
Author Response
Dear Reviewer,
We thank you for taking the time for reviewing our manuscript and appreciate your timely feedback. Please find the responses to your valuable comments below. We remain at your disposition for further questions or concerns.
Major points:
- The authors based their results in western blot detection. Western blot is not a validated test for nodal/paranodal antibody screening. Furthermore, it is recommended to perform at least two techniques to confirm the positivity, because false positives have been previously described (Martín-Aguilar et al, Neurology, 2020). The recent published guidelines on diagnosis and treatment of CIDP (Van den Bergh et al. EAN/PNS Guidelines, J Peripher Nerv Syst,2021) recommend performing a cell-based assay (CBA), ELISA or teased-nerve immunohistochemistry and always performing a confirmatory test with a second technique. In fact, the authors discussed that antibodies detected by western blot may reflect non-specific bindings without pathological relevance (discussion, line 239), but they also state that two articles demonstrated the equivalence between ELISA and CBA. These two articles demonstrate that ELISA and CBA are equal methods to detect nodo-paranodal antibodies, but not western blot. Thus, I recommend performing at least two different techniques to the western blot to confirm the positive patients in your case series.
- Response: While we are cognisant of this limitation, the western blot technique was the only methodology available commercially. We fully agree that these positive findings are not final. Most recently, Mayo added a cell-based assay for anti-neurofascin 155 IgG4.
- Response: Regarding the articles comparing different assay techniques:
- Articles 6 and 15 demonstrated equivalence between CBA and ELISA. Non-specific binding is another caveat of the ELISA techniques, something that was exhaustively described in the MOGAD literature as earlier studies demonstrated the MOG antibody to exist in individuals with MS when western blot and ELISA were used as assays
- Article 24 by Kodoya et al, demonstrated equivalence between western blot, CBA and Immunohistochemistry techniques
- In the introduction the authors state: “Anti-neurofascin-antibodies (ANFAs) might characterize an emerging group of immune-mediated neurological disorders that have not yet been fully described in terms of presentation, diagnosis, and management.” This affirmation needs clarification: in terms of peripheral nervous system, anti-neurofascin antibodies are disease-specific antibodies that have been described in a small subset of patients with CIDP sharing immunopathologic mechanisms, clinical features, and treatment response and differing from those of typical CIDP.
- Response: We have modified our introduction and attempted to soften the association of ANFAs with common immune-mediated CNS disorders.
- Moreover, this has led to the appearance of the autoimmune nodopathy (AN) diagnostic category in the recent update of the European Academy of Neurology/ Peripheral Nerve Society CIDP diagnostic guidelines (Van den Bergh et al. EAN/PNS Guidelines, J Peripher Nerv Syst,2021). On the other side, combined central and peripheral demyelination (CCPD) has been described in two cohorts: Kawamura et al. Neurology 2013 and Cortese et al., J Neurol Sci, 2016, last reference not included in the manuscript. Please, add a better explanation of neurofascin antibodies in peripheral nervous system and discuss that its value in CNS and in CCPD is not so well-known.
- Response: we have modified the introduction to reflect a better explanation of neurofascin antibodies in the peripheral nervous system. Additional information on the role of neurofascin antibodies in a rat model of acute inflammatory demyelinating polyradiculopathy was included in the discussion.
- Response: As suggested, the article by Cortese et al is added now: thank you for the valuable suggestion!
- From table 1, 5 out of 6 patients have been tested for MOG/NMO antibodies. Which test have you used to test NMO/MOG antibodies? Patient 1 should be also tested for NMO/MOG, as she has a longitudinally extensive myelitis and NF155 have been reported to appear in some NMO patients (Chang et al. Clin Exp Immunol, 2021).
- Response: The Fluorescence-Activated Cell Sorting (FACS) Assay by the Mayo Clinic Laboratory was used to test NMO/MOG antibodies. We agree that patient should have also been tested for NMO/MOG antibodies but after some initial testing, she declined further workup and asked for comfort care. She passed away peacefully few days later.
Minor points:
- Final diagnosis is not clear in most of presented cases and a lot of tests and explanations for each case can be collected in a table. Consider reducing the length of the clinical cases, trying to include the results of the tests in table 2 and, in each case, try to approximate the diagnosis whenever possible.
- Response: As kindly suggested, we have tried to approximate the diagnosis, and this is now reflected in the new version of the manuscript.
- Response: In the first manuscript draft, we tried to shorten the length of the clinical cases all while trying to keep important clinical information that might be of interest to clinicians. To that purpose, we have included table 1 summarizing all normal tests and highlighted abnormal ones in the last column.
Reviewer 4 Report
This short series reports 6 patients suffering from central nervous system involvement probably related to anti-neurofascin antibodies (ANFA).
Cases were described a satisfying amount of details. Discussion is clearly presented and provides a panorama of the question.
As authors stated, it remains uncertain if ANFA-associated syndromes are real association or fortuitous findings. According to the rarity of available cases in the literature, it seems interesting to publish this short series.
* Did authors followed other ANFA-positive patients, especially displaying lesions of the peripheral nervous system ?
* How long did it take ti gather this series? Follow-up was very short, does it mean that patients were all recently diagnosed as part of an effort to generalize ANFA test?
* why did you test ANFA especially in these patients ? Do you check ANFA in every atypical central or peripheral synromes?
* Did you test some patients’ CSF for ANFA?
* Case 1: According to MRI results showing a V-sign, did you test B12 vitamin, and homocystein? We assume that nitric oxid acute intoxication was ruled out.
* Case 2: no pleocytosis but high neutrophils and eosinophils count (p2L60): do you mean that the ratio of floating cells in CSF was different from the ratio expected from blood? or do you mean an abnormal CSF count?
* Case 3: Figure depicting example of brain lesions could be interesting.
* Table 2 is not clearly tabulated : would it be possible to expand columns and may be to abbreviate results?
*Ref 26 should be corrected
* Minor corrections: check text for double spaces.
Author Response
Dear Reviewer,
We thank you for taking the time for reviewing our manuscript and appreciate your timely feedback. Please find our responses to your valuable comments below. We remain at your disposition for further questions or concerns.
* Did authors followed other ANFA-positive patients, especially displaying lesions of the peripheral nervous system?
Response: These patients were evaluated in the MS/Neuroimmunology clinic where patients with central nervous system diseases are exclusively referred. Our overall goal was to describe patients with central nervous system lesions who tested positive for ANFAs. We tested for ANFAs in patients with atypical clinical findings as detailed in the discussion and the patients that tested positive were included in the case series. Patients with positive ANFAs and peripheral nervous system diseases are evaluated and followed by our neuromuscular colleagues.
* How long did it take this gather this series? Follow-up was very short; does it mean that patients were all recently diagnosed as part of an effort to generalize ANFA test?
Response: Indeed, the follow-up was short. We started testing for this entity in the last 2 years in patients who had combined CNS and PNS disease or individual with a clinical and MRI highly suggestive of MS with some atypical features such as active disease in older patients, atypical CSF findings, etc. as outlined in the discussion. As the test is available commercially, our goal was not to generalize the ANFAs testing but an attempt to explain some of the atypical findings. We understand that these antibodies might be an epiphenomenon or of dubious significance and might not be diagnostic but believe that these patients should be kept under the clinician radar for further testing in the future because of the presence of ANFAs.
* Why did you test ANFA especially in these patients? Do you check ANFA in every atypical central or peripheral syndromes?
Response: We tested ANFAs in these patients due to presence of atypical clinical features, imaging and laboratory data. Patient 1 had AMSAN and the cervical spine MRI showed a dorsal column lesion due to which we suspected a common pathophysiologic mechanism affecting the PNS and CNS, and hence we tested for ANFAs given their association with central and peripheral demyelination. Patient 2 had eosinophils on CSF analysis due to which we suspected that the pathology was mediated through a complement-dependent process rather than T-cell mediated process, and hence we checked for ANFAs as other commonly present antibodies in CNS demyelination were ruled out. Patient 3 ‘s clinical presentation was atypical for MS but the MRI was convincing for demyelinating lesion. Given her older age of presentation, we suspected an alternate pathology and hence, we checked for ANFAs. Patients 4 had demyelinating lesions typical for MS but given her older age of presentation, we checked for ANFA to evaluate for alternative pathology. For patient 5, older age of presentation and presence of cervical peripheral nerve root enhancement made us suspicious for alternative pathology and hence, we checked for ANFA. Patient 6 had events of PNS and CNS involvement that were separated in time, and hence, we checked for ANFA to evaluate for alternative pathology.
We focused on patients with atypical central nervous system syndromes since these patients were evaluated in the MS/neuroimmunology clinic where the referral population mainly consists of central nervous system lesions. We did not evaluate patients with atypical peripheral syndrome which is a limitation of our study and has been updated in the manuscript (Section, “Discussion”, page 3 of 17, lines 281-286).
* Did you test some patients’ CSF for ANFA?
Response: Thank you for bringing this to our attention. We thought about checking ANFA in patients’ CSF since we were evaluating patients with central nervous system pathology but we did not find a commercially available test. It is probably performed in research labs but since there was no commercially available test, we did not check for it which is a limitation of our study. We have updated this in the manuscript (Section, “Discussion”, page 3 of 17, lines 286-289). Additionally, as stated in our discussion Stitch et al., reference 10, assayed the simultaneous presence of serum and CSF ANFAs in five patients with MS, and none had ANFAs in the CSF. Future studies will elucidate which test, serum or CSF analysis, might have a better sensitivity for ANFAs.
* Case 1: According to MRI results showing a V-sign, did you test B12 vitamin, and homocysteine? We assume that nitric oxid acute intoxication was ruled out.
Response: We tested for vitamin B12 and homocysteine and these were unremarkable. There was no clinical evidence to suggest nitric oxide exposure. This has been updated in table 1 in the manuscript, and in the description of case I (lines 29030).
* Case 2: no pleocytosis but high neutrophils and eosinophils count (p2L60): do you mean that the ratio of floating cells in CSF was different from the ratio expected from blood? or do you mean an abnormal CSF count?
Response: We meant abnormal CSF count in this case.
* Case 3: Figure depicting example of brain lesions could be interesting.
Response: Some brain MRI images of patient 3 were added.
* Table 2 is not clearly tabulated: would it be possible to expand columns and may be to abbreviate results?
Response: Table 2 has been revised and updated.
*Ref 26 should be corrected
Response: References were corrected.
* Minor corrections: check text for double spaces.
Response: Done and thanks again for taking the time to provide us with feedback.
Round 2
Reviewer 2 Report
The authors have appropriately revised the manuscript and have highlighted the limitations of the studies.
Author Response
Dear Reviewer,
We thank you for taking the time to review our manuscript and appreciate your timely feedback. We remain at your disposal for further questions or concerns.
Best,
Sincerely,
Navnika Gupta
Reviewer 3 Report
The article has improved in its form, but the main limitation has not been adressed. I still have important concerns about the methodology of antibody detection and I cannot accept the manuscript if there is no confirmatory technique, as the EAN/PNS guidelines recommend.
The results are based in a detection method that is not validated. This reviewer is fully aware that there is no validated commercial kit for neurofascin antibodies and this is the reason why you should send the samples to a expert lab that perform the proper techniques for neurofascin antibodies detection (CBA, ELISA or teased-nerve fibers) or try to perform them in your lab using the correct publicated methods.
Author Response
Dear Reviewer,
We thank you for taking the time one more time to review our manuscript, and we appreciate your timely feedback. Please find the responses to your valuable comment below. We remain at your disposal for further questions or concerns.
The article has improved in its form, but the main limitation has not been addressed. I still have important concerns about the methodology of antibody detection, and I cannot accept the manuscript if there is no confirmatory technique, as the EAN/PNS guidelines recommend. The results are based on a detection method that is not validated. This reviewer is fully aware that there is no validated commercial kit for neurofascin antibodies, and this is the reason why you should send the samples to an expert lab that performs the proper techniques for neurofascin antibodies detection (CBA, ELISA, or teased-nerve fibers) or try to perform them in your lab using the correct publicated methods.
Response: Thank you for the response. We agree with the reviewer that, in an ideal setting, antibody testing should have been performed by two techniques and the lack thereof is a limitation of our study. This has been updated in the manuscript.
We contacted the neuromuscular lab at Washington University School of Medicine regarding the neurofascin antibodies testing methodology. We were told that when the test was originally launched, it was done by ELISA. However, the Western Blot methodology was found to be more specific. Additionally, the test has been validated. We recognize that the gold standard methodology is a subject of debate and believe that as science evolves, this question will be clarified, and more commercial tests will be available.
Best,
Navnika Gupta, MD